# Reversed Polarity bi-tDCS over M1 during a Five Days Motor Task Training Did Not Influence Motor Learning. A Triple-Blind Clinical Trial

**DOI:** 10.3390/brainsci11060691

**Published:** 2021-05-25

**Authors:** Laura Flix-Díez, Miguel Delicado-Miralles, Francisco Gurdiel-Álvarez, Enrique Velasco, María Galán-Calle, Sergio Lerma Lara

**Affiliations:** 1Department of Physical Therapy, University of Valencia (UV), 46003 Valencia, Spain; lauflix@alumni.uv.es; 2Instituto de Neurociencias de Alicante (UMH-CSIC), 03550 Sant Joant d’Alacant, Spain; mdelicado@umh.es (M.D.-M.); e.velasco@umh.es (E.V.); 3Department of Physical Therapy, Occupational Therapy, Rehabilitation and Physical Medicine University of Rey Juan Carlos, 28922 Alcorcón, Spain; f.gurdiel.2019@alumnos.urjc.es; 4Health Sciences Faculty, Centro Superior de Estudios Universitarios La Salle, Universidad Autónoma de Madrid, 28023 Madrid, Spain; mariagalanc9@gmail.com; 5Motion in Brains Research Group, Centro Superior de Estudios Universitarios La Salle, Universidad Autónoma de Madrid, 28023 Madrid, Spain

**Keywords:** motor learning, bilateral transcranial direct current stimulation (bi-tDCS), motor training, motor hand dexterity, healthy subjects, somatosensory system

## Abstract

Transcranial direct current stimulation (tDCS) has been investigated as a way of improving motor learning. Our purpose was to explore the reversal bilateral tDCS effects on manual dexterity training, during five days, with the retention component measured after 5 days to determine whether somatosensory effects were produced. In this randomized, triple-blind clinical trial, 28 healthy subjects (14 women) were recruited and randomized into tDCS and placebo groups, although only 23 participants (13 women) finished the complete protocol. Participants received the real or placebo treatment during five consecutive days, while performing a motor dexterity training program of 20 min. The motor dexterity and the sensitivity of the hand were assessed pre- and post-day 1, post 5 days of training, and 5 days after training concluded. Training improved motor dexterity, but tDCS only produced a tendency to improve retention. The intervention did not produce changes in the somatosensory variables assessed. Thus, reversal bi-tDCS had no effects during motor learning on healthy subjects, but it could favor the retention of the motor skills acquired. These results do not support the cooperative inter-hemispheric model.

## 1. Introduction

Motor dexterity is defined as the ability to accurately manipulate small objects during a task with fine movements [1,2]. Manual dexterity is probably one of the most developed among human skills, and has a vital role, being necessary to perform the basic activities of daily living. When manual dexterity is impaired, either due to pathological conditions or aging, the impact in the quality of life of patients is quite high [3]. As an example, in US and in Europe, stroke patients need human assistance for the activities of daily life, plus the costs of the rehabilitation needed to recover this dexterity, where the burden of this pathology alone costs USD 34 billion a year [4] and EUR 60 billion in 2017 [5].

The clinical practice guidelines and other research works support the rehabilitation motor training as the most accepted strategy for stroke patients [6,7,8]. In consequence, rehabilitation and neuroscience fields are putting great efforts into the development of treatments focused on the recovery of hand motor functions. The physiological basis that underlies motor training is the plasticity of the central nervous system (CNS). This plasticity refers to the changes in the motor circuitry, in connectivity, synaptic strength, or neuronal excitability as a consequence of training. Therefore, motor skill learning is the speed or accuracy improvement mediated by training [9,10]. The primary motor cortex (M1) has an important role in this learning [11], in fact, motor dexterity improvement is associated with plasticity in M1, specifically as an increase in the functional representation of the trained muscles [9]. Due to the close relation between the motor and sensory cortex, motor learning has an effect on sensory perception, producing changes related with somatosensory input or movement perception [12]. Moreover, motor learning has a characteristic time course: within-session learning is observable in the first training session and across session learning is only observable after multiple training sessions [9].

In recent years, complementary to rehabilitation based on motor training and aiming to maximize its effects, non-invasive brain stimulation (NIBS) tools have gained attention [13,14]. Among them, transcranial electrical stimulation has recently gained popularity in human neuroscience research [15]. Specifically, transcranial direct current stimulation (tDCS) is the most studied of the non-invasive techniques in this field. It consists of the application of weak intensity currents (1 or 2 mA), through two or more electrodes placed on the scalp, for 5–30 min [15,16]. There is evidence that tDCS over M1 can improve motor learning (within- and across-sessions) and consolidation of motor skills in healthy controls and patients with motor impairments [17,18]. Similar effects are produced over fine motor skills [19]. Supporting these behavioral results, tDCS intervention over M1 can modify cortico-motor excitability [13,20]. Subsequently, the proposed physiological basis supporting tDCS effects is, similar to the motor training, changes in the motor circuitry [2]. 

The tDCS effects seem dependent on the polarity of the electrodes: it is suggested that anodal stimulation depolarizes and cathodal stimulation hyperpolarizes the neurons [20,21,22]. The anodal-tDCS (a-tDCS, Figure 1a) over contralateral M1 (1 mA and 20 min), which increases excitability, is the most studied application for motor learning studies [23]. On the contrary, the cathodal tDCS over contralateral M1 (c-tDCS, Figure 1b), which decreases excitability, seems to limit motor learning [21,24] or not influence it [17,25]. On the other hand, cathodal tDCS over the ipsilateral M1 seems to facilitate motor learning [14,26], probably due to the interhemispheric inhibition phenomenon found in the human motor cortex [27]. Then, searching new and more efficient tDCS protocols, bi-hemispheric tDCS application (anode on contralateral M1 and cathode on ipsilateral M1, bi-tDCS, Figure 1c) was proposed, seeming to have greater effects than unilateral a-tDCS stimulation in healthy subjects [14,28,29,30,31,32]. The reasoning for this tDCS application is guided by an interhemispheric competition model, in which both hemispheres suppress each other through inhibitory connections [33]. Although, Waters et al. [32] found that the bi-hemispheric tDCS produced similar improvements in motor learning regardless of the polarity of the electrodes. In the scope of this result, some authors proposed a different model: interhemispheric cooperation. However, it was described that the tDCS effects can be reversed with higher intensity, or a longer duration of stimulation, than 1 mA or 15–20 min [34,35]. Waters et al. used 2 mA during 25 min, thus it is plausible that the reversed-polarity (RP) bi-hemispheric tDCS (bi-RP-tDCS, Figure 1d) effects they found differ from the canonical modulation of excitability [36]. This lack of understanding of the biological consequences [2], and the lack of consensus in the protocol of action [5,7], are two major limitations for the generalization of this technique. Therefore, the main purpose of this work is to explore the bi-RP-tDCS application effects with standard parameters (1 mA and 20 min) on uni- and bimanual motor dexterity, along the three temporal components of motor learning (within-session, across-session, and retention) during five days of consecutive unimanual dexterity training. In addition, due to the sensory changes produced as part of motor learning [12], we also explored the effects of the interventions in somatosensory variables. Secondarily, we explored whether manual dexterity training and the bi-RP-tDCS intervention could modify grip strength. This work could clarify whether the bi-RP-tDCS enhances motor learning or not, giving more weight to the competitive model, which could more adequately explain brain performance.

## 2. Materials and Methods

### 2.1. Study Design 

This study was a randomized triple-blind clinical trial controlled with a placebo group. 

The participants were assigned equally (1:1) into two groups, bi-RP-tDCS or placebo, by a permuted block randomization (GraphPad Software).

### 2.2. Participants

Twenty-eight healthy volunteers (14 women), with a mean age and standard deviation of 25.36 ± 4.45, participated in this study. The inclusion criteria was being right handed, healthy and young (18–35 years), because performance and motor learning may be affected by age [18]. Participants filled out the Edinburgh Inventory to determine their dominant hand [12]. The participants met the NIBS criteria [16], thus the exclusion criteria was: metal or skin lesions on the head, brain stimulation in the last 6 months, family history of epilepsy or seizures, and a pacemaker or any cardiac involvement. In addition, we also considered the inability to move or alterations in the fingers, hands or wrists as a motive for exclusion, alongside the inability to understand or execute the task, drug consumption that may influence cognition (according to the WHO, harmful alcohol consumption is considered to be the intake of more than 40 g/day in women and more than 60 g/day in men [37]), and pregnancy.

All study data was collected in the CSEU La Salle-Faculty of Health Sciences, in Madrid. The study was accepted by the GAE Clinical Research and Ethics Committee of the Hospital Infantil Universitario Niño Jesús (R−0022/18). Before the intervention, all subjects signed an informed consent in accordance with the Declaration of Helsinki (World Medical Association, 2002). The protocol was preregistered on clinicaltrials.gov with the NCT03931512 number. 

### 2.3. Intervention

#### 2.3.1. Motor Dexterity Training

An exercise program for the dominant hand was designed with the Purdue Pegboard Test (PPT). The program consisted of a sequence of six exercises (Figure 2a) and the participants repeated this sequence during the 20 min of intervention. All exercises were performed with the dominant hand.

There is evidence that supports the generation of a more effective performance and learning that appears when the attentional focus is directed to the effects of the movement of the individual on his environment. That is, to induce an “external approach” in comparison to an “internal approach” [38]. For this reason, the exercise program focused the patient’s attention on placing the sticks in their corresponding holes and not on the sensation of the movement of the hand when performing the training. To encourage the subjects, they were instructed to do it as quickly as possible, trying to improve the number of exercises they could complete with respect to the previous day.

#### 2.3.2. Transcranial Direct Current Stimulation

The tDCS was applied with a multichannel wireless transcranial current stimulator (Starstim tCS^®^, Neuroelectrics^®^, Barcelona, Spain) and programmed with the NIC 2.0 software (Neuroelectrics^®^, Barcelona, Spain). A bi-RP-hemispheric configuration was applied [32]. Electrodes were placed based on the international electrode placement 10–20 system, with a non-conductive neoprene cap, sized M (Neuroelectrics^®^ Neoprene Head cap) (Figure 1b) [16]. The application of bi-RP-tDCS was simultaneously applied with motor training, as in other works [39,40,41], and during five consecutive daily sessions of 20 min, since this was the application duration that most studies used [19] (Figure 2b,c). In the intervention group, a density of 0.04 mA/cm^2^ was obtained from 2 circular sponge electrodes (25 cm^2^) and a current intensity of 1 mA. An initial and final ramp of 10 s were programmed, the most used form so far (Figure 2b) [16,23,42].

#### 2.3.3. Placebo and Blinding Procedure

The placebo group received a validated protocol in order to blind the participants [42]. The current was programmed to increase intensity during 10 s, until it reached the application intensity (1 mA) and, immediately, a decrease ramp started, lasting 20 s to reach zero intensity. This left a total stimulation time of 30 s and left the rest of the session without current (19:30 min) (Figure 2b). This protocol often evokes the tingling or itching perception as tDCS does, making the subjects unable to distinguish between treatments, especially in independent groups designs like this one [16,42].

In order to blind the participating researchers, we used a coding system to mask the patient allocation and the intervention application. The therapist codified the tDCS and placebo intervention in numbers (0 and 1, respectively), and programmed the intervention protocols on the software device under the number tag. The evaluator again codified both codified groups (0 and 1) with letters (A and B, respectively). The evaluator also performed the random allocation of the participants in both groups (A and B) and changed the protocol names in the software device. Finally, the statistician was also blinded and performed the analysis under the A and B names. Then, the participants, the therapist, the evaluator, and the statistician were blinded. The blinding was maintained until the study was completed and the data set blocked [43].

In addition, to assure the blinding, other aspects of study design were customized. The stimulator was programmed to never show the current intensity administered, only showing the remaining time and a categorical indication of impedance between the electrodes and the scalp indicating the quality of contact (optimal, moderate, or bad). The feedback that the patients received, observing the quality contact indication, seemed to reduce the risk of unmasking [44] and, in case of low-quality contact, the therapist could improve the contact, adding saline solution to the electrodes or removing the hair under the electrodes.

### 2.4. Measurement Protocol

When each participant read and signed the informed consent, some sociodemographic data were collected from participants: age, sex, level of education, personal and family history, and drug intake. 

For the assessment of within-session, across-session, and the retention components of motor skill learning, we performed four measurements over ten days (Figure 2c). The variables assessed in the measurements were always performed in the same order: somatosensory, grip strength, and motor dexterity. Finally, to reduce variability and avoid bias, all variables were assessed by the same researcher three times and averaged.

#### 2.4.1. Motor Dexterity Variables 

The main variable was the fine unimanual motor dexterity, measured by the standardized Purdue Pegboard Test (PPT), (Lafayette Instrument, San Diego, CA, USA, Model 32020). It is commonly used to clinically evaluate the function of the hand [45]. The test consisted of taking sticks from the nearest bowl with the index finger and thumb of the dominant hand, one by one, and placing them as quickly as possible on the ipsilateral column. The score was the number of sticks placed for 30 s. One of the exercises performed during training was doing this action in a very similar way (Figure 2, Exercise 1).

We also assessed the bimanual fine motor dexterity, through the bimanual test from the PPT. The task was similar to the unimanual motor dexterity, but with both hands taking and placing sticks in their respective columns at the same time. Here, the score was the number of completed horizontal lines in 30 s.

On the other hand, uni- and bimanual gross motor dexterity were measured by the Minnesota Manual Dexterity Test (MMDT), through the placing and the turning test, respectively. Both tests are well known and used in rehabilitation [46,47]. In general, the MMDT consists of 60 black cylinders, in which one side is painted in red, which fit in a black board with a hole pattern of 4 rows and 15 columns. Both tests consisted of taking cylinders previously ordered (4 × 15) and placing them in the board in a specific way, order, and in the shortest time possible (seconds). The placing test (unimanual), consisted of taking the cylinders from outside the board and placing them in it with the dominant hand. The sequence started from the lower ordered line, taking first the cylinder on the side of the dominant hand and placing it in the same side, but in the top line of the board. The sequence continued across the cylinders of the lower line and when it was finished, subjects continued consecutively with the superior line to reach the top line. This placement was performed like a mirror. The Turning test (bimanual) consisted of taking the cylinder from the board with one hand, transferring the cylinder to the other hand and placing it again on the same board place, but showing the other side of the cylinder (different color). The sequence started from the upper line on the dominant hand side. When each line of cylinders was placed, the role of the hands was inverted. For example, a right-handed subject would take the cylinders of lines 1 and 3 with the right hand and lines 2 and 4 with the left hand.

In all motor tests, a “warming” attempt was allowed and subjects were instructed to perform the test as fast as possible. Moreover, after training, in order to avoid the possible fatigue or the “warm-up” effect [9], all the motor variables were assessed 30 min after the training finished [48].

#### 2.4.2. Somatosensory Variables 

All somatosensory variables were assessed in the dominant hand, with the subject sitting comfortably and with the hand resting on the table. A sheet was used to avoid the participant seeing the measurements.

The mechanical detection threshold was assessed on the thenar eminence using Von Frey Monofilaments [49]. Ordered by increasing size, different monofilaments touched the skin of the subjects while the participant was asked, in each stimulus, to confirm whether he had noticed it or not. Once the threshold was detected, the caliber of the last filament was recorded.

The pressure pain threshold was assessed in the thenar eminence and the dorsal area of the diaphysis of the second metacarpal bone with a handheld digital algometer (FDX-25 Wagner Instruments, Greenwich, CT, USA). Both measures were done due to the different thresholds found on muscle and bone [50,51]. The rubber tip of the algometer was applied perpendicularly to the skin of the participant, and the force was gradually increased at a rate of approximately 1 kg per second. To ensure a constant gradual increase in force, the therapist underwent previous training and used a metronome at 60 beats per minute as a guide (ppm) [52]. The participants were instructed to give a verbal command to stop when the feeling of pressure turned into pain, and the pressure applied at this moment was recorded.

The two-point discrimination threshold was measured in the thenar eminence using esthesiometry [53] in thenar eminence. The tips of the esthesiometer were applied, gradually increasing the separation between them. At the same time, the participant was required to answer if he/she felt one or two points in each stimulus. Once the participant was capable of discriminating two points, the distance between tips was recorded.

#### 2.4.3. Grip Strength

The grip strength was assessed in the dominant hand using the Jamar analog hand dynamometer (Saehan Hydraulic Hand Dynamometer, 0–90 kg). Subjects kept the arm close to the trunk, with the elbow flexed at 90° and the forearm in a neutral prone-supination [51]. They were instructed to perform a grip as intense as possible. The average of three grip force measurements was obtained following the current recommendations [52].

#### 2.4.4. Physical Activity and Sleep Quality 

Physical activity level and sleep quality of the subjects were assessed in a self-administered way with the International Physical Activity Questionnaire (IPAQ) (Long version: USA Spanish version 3/2003) [53] and the Pittsburgh Sleep Quality Index (PSQI) [54]). The IPAQ presents different blocks of questions related to physical activities performed in the different domains of daily life, scoring the activity level in MET-minutes/week. The PSQI is one of the most recommended and used instruments in this field as it is brief, simple, and well accepted by patients. It consists of 19 items that analyze different determinants of sleep quality grouped into seven components: quality, latency, duration, efficiency and sleep disturbances, use of medication for sleep, and diurnal dysfunction [55]. The total score of the PSQI is between 0 and 21, with a higher score denoting worst sleep quality. Subjects with more than 5 points are considered poor sleepers. These two variables were measured on the last day of the protocol due to its retrospective character.

### 2.5. Sample Size

The study sample size was estimated based on the work of Waters et al. [32]. We used G*power 3.1 14 software [56]. To achieve statistical significance, with an effect size of 1.3 (Cohen’s d) from the main variable (unimanual dexterity task), a power of 0.8 and an α-error of 0.05, the minimum number of total participants was established in *n* = 11 per group (22 total subjects). However, to account for possible dropouts during the consecutive training and the retention period, we added 25–30% of subjects to the sample (+6), reaching a recruited sample of 28 subjects.

### 2.6. Statistical Analysis 

All statistical analyses were performed using IBM SPSS Statistics for Windows (Version 26.0) Armonk, NY: IBM Corp. Data distribution was studied with the Shapiro-Wilk test. For descriptive data, we compared quantitative and qualitative variables with independent Student’s *t*-test and x^2^ test, respectively. Regarding variables that did not accomplish normality, respective nonparametric tests were performed.

To explore training and bi-RP-tDCS effects on each group, we used two-way repeated measures ANOVA (mixed model, RM ANOVA), focusing on the time factor (basal conditions, post-one session, post-five sessions and after 5 days without training), and the time*group factor (bi-RP-tDCS vs. placebo) interaction. We used post-hoc comparisons in a deeper analysis to evaluate specific temporal components of motor skill learning. We performed exploratory post-hoc for ANOVA with a *p* < 0.1 because of the rather small sample size. Complementary to the ANOVA analysis, we performed, in the Appendix A, a linear mixed model in order to better describe the time, group, and time*group interaction effects.

The dropouts were included in the analysis (intention to treat analysis). The threshold for statistical significance was *p* < 0.05.

In case there were differences in demographics data between groups, we performed a linear regression model in order to assess the possible influence over motor learning.

## 3. Results

Twenty-eight participants were recruited, and their descriptive data is indicated in Table 1. We observed in the enrollment procedure (Figure 3) that we had five dropouts and the descriptive data of that sample is shown in the Appendix A.

### 3.1. Bi-RP-tDCS Effects on Motor Learning Training

#### 3.1.1. Unimanual Fine Motor Dexterity

The main variable of this study was the unimanual fine motor dexterity, which was specifically trained during the intervention. Training improved motor dexterity in both groups as there was a statistically significant improvement over time (Table 2). To specifically explore which learning temporal components were improved, we assessed the within-session learning comparing the motor dexterity in basal conditions and after the first training session. This component was significantly improved in both the bi-RP-tDCS and placebo groups (*p* < 0.001, paired *t*-test Cohen’s d = 1.36 and 1.26, *n* = 15 and *n* = 13, respectively). To analyze the across-session component, we compared the basal conditions and after finishing the five days training data, which was also improved in both groups (*p* < 0.001, paired *t*-test, Cohen’s d = 2.62, *n* = 13 for bi-RP-tDCS and Cohen’s d = 1.87, *n* = 11 for placebo). Finally, in order to assess the retention component, the ability to maintain the dexterity level after five days of non-training, we analyzed the difference of motor dexterity after being finished all training and after five days without training. Here, we observed a non-significant tendency in the placebo group to experience a decrease in their motor dexterity level (*p* = 0.093, paired *t*-test, Cohen’s d = 0.28, *n* = 11), not presented in the bi-RP-tDCS group, which maintained a skill level similar to the end of training level (*p* = 0.480, paired *t*-test, Cohen’s d = 0.15, *n* = 12) (Figure 4a).

Regarding the bi-RP-tDCS effects, we analyzed the time * group interaction, which only showed a small but not significant effect (Table 2). We performed the post-hoc in order to analyze the temporal components in an exploratory way. We found that manual dexterity was similar in both groups at basal conditions (*p* = 0.275, independent *t*-test and Cohen’s d = 0.15, *n* = 28). The motor dexterity level post first and post 5 days training were similar between groups (*p* = 0.681, and 0.117, independent *t*-test, Cohen’s d = 0.16 and 0.57, *n* = 28 and 24, respectively). Finally, for the retention measure, when both groups were compared, although non-significant, the means were also different between groups (*p* = 0.06, independent *t*-test, *n* = 23, Cohen’s d = 0.83) (Figure 4a). 

In summary, motor training improved motor learning in both groups, while bi-RP-tDCS had no strong effects, only showing a not statistically significant tendency to favor the retention component.

#### 3.1.2. Bimanual Fine Motor Dexterity

For the evaluation of the possible bilateral effects of unimanual training, we assessed bimanual fine motor dexterity (Table 2). Training improved this motor learning over time, specifically both the bi-RP-tDCS and the placebo group improved their bimanual fine dexterity in the within- (*p* < 0.05 and <0.001, Cohen’s d = 0.71, *n* = 15 and 13, respectively) and across-session components of motor learning (*p* < 0.001, Cohen’s d = 1.17 and 1.04, *n* = 13 and 11) and showed similar learning retention (*p* > 0.05, Cohen’s d = 0.32 and 0.04, *n* = 12 and 11). Analyzing the intervention factor, we had no significant effects (Figure 4b). Unimanual dexterity training improved the performance in bilateral motor dexterity tasks, although bi-RP-tDCS had no effect over this learning.

#### 3.1.3. Uni- and Bimanual Gross Motor Dexterity

To assess transference of motor learning between modalities, we measured other motor hand tasks, such as uni- and bimanual gross dexterity. Similar to fine motor dexterity, both gross motor tasks improved with training (Table 2), improving their uni- and bimanual gross dexterity in the within- (*p* < 0.01, Cohen’s d = 0.81 and *n* = 13 for bi-RP-tDCS group and *p* < 0.001, Cohen’s d = 0.56, and *n* = 12 for placebo group) and across-session components of the motor learning (*p* < 0.001, Cohen’s d = 1.59 and 1.18, *n* = 11 and 10, respectively). They also showed similar task retention (*p* > 0.05, Cohen’s d = 0.04 and 0.31, *n* = 10, respectively). 

The bi-RP-tDCS had no effects on bimanual gross dexterity, although seemed to show a non-significant tendency to favor motor learning of unimanual gross dexterity (Table 2). We analyzed the temporal components in an exploratory way. Regarding the differences between groups at specific times, unimanual gross dexterity learning was similar at basal, post first training, and retention conditions (*p* > 0.05, Cohen’s d = 0.22, 0.57 and 0.38, *n* = 25, and 20, respectively). The unimanual gross dexterity at post-five days training was the only temporal point that showed a non-significant tendency to increase in the bi-RP-tDCS group (*p* = 0.069, Cohen’s d = 0.77, *n* = 21) (Figure 4c). Referring to bimanual gross motor dexterity, a constant tendency to show better results in the bi-RP-tDCS group was also observed, but the RM ANOVA showed a practically null effect from the intervention itself (*p* = 0.995). Therefore, we suggest that this tendency to be different is due to the basal differences between groups (Figure 4d), where the bi-RP-tDCS group tended to be higher than the placebo (*p* = 0.06, *n* = 25, Cohen’s d = 0.78).

Thus, motor training improved non-trained uni- and bimanual gross dexterity tasks and bi-RP-tDCS had unremarkable effects over this learning, only showing a non-statically significant tendency to increase the across-sessions component of learning on the unimanual task. The same effects were found for all motor variables when only the sample that completed the study was analyzed, excepting the improvement of the gross unimanual task on the within-component of the tDCS group compared to the placebo (*p* < 0.05, Appendix A).

#### 3.1.4. Sleep Quality Did Not Influence Motor Learning

We found a significant difference in the sleep quality between groups (Table 1). The tDCS group were ‘’better sleepers’’, which could influence motor learning. However, we performed a linear regression and the sleep did not predict motor learning (dependent variable: difference between post 5 days training and basal condition, *R*^2^ = 0.074, *p* = 0.232). Moreover, we compared the proportion of bad sleepers between groups and there were no differences (*p* = 0.123). Thus, the sleep quality did not affect motor learning.

### 3.2. Bi-RP-tDCS Effects on Grip Strength

We also explored the effect of motor training and bi-RP-tDCS on grip strength, finding that training changed grip strength but not bi-RP-tDCS (Table 3). Thus, training only produced a slight and transient improvement in the strength of the placebo group, and the bi-RP-tDCS application produced no effects (Figure 5). The same was found when the complete sample was analyzed (Appendix A).

### 3.3. Bi-RP-tDCS Effects on Somatosensory Variables

Finally, we explored whether the training and bi-RP-tDCS application produced any change over somatosensory variables. We only found that motor training changed the pressure pain threshold in the diaphysis of the second metacarpal bone, but the bi-RP-tDCS had no effect (Table 4). In conclusion, as we found with grip strength, motor training and bi-RP-tDCS did not produce relevant sensory changes (Figure 6a–d). The same was found when the complete sample was analyzed (Appendix A).

## 4. Discussion

The main aim of the present work was to explore bi-RP-tDCS application effects over motor learning, combined with a five-day training program. The hypotheses supported that the application of bi-RP-tDCS does not improve motor learning as other application modes like anodal or bi-hemispheric tDCS do when used following similar parameters [14,28,29,30,31,32]. These results provide information on the direction of an interhemispheric competition model [33]. We found that the training program produced motor learning in both groups and, while the bi-RP-tDCS application did not facilitate motor learning acquisition, it could enhance the retention of the motor skill trained. 

Motor learning is produced in both groups because they showed a substantial improvement in their unimanual fine dexterity in the PPT task. In addition, this improvement was also found in non-trained motor dexterity tasks (transference). Said effect, although sometimes reduced, was maintained after five days without training (retention). Transference and retention are intrinsic features of motor learning [38], therefore supporting that our training program produced motor learning. This is further confirmed by similarities with other works that use PPT for motor training and finding performance improvement and plastic changes in M1, such as an increase in the corticospinal excitability [57]. Although, we cannot discard the possible learning produced by the repetition of the tasks during the assessment itself as having contributed to transference and retention.

While our results do not support the idea that bi-RP-tDCS application will have robust effects on motor learning, it is well accepted that the a-tDCS on M1 contralateral improves unimanual motor learning [29]. For example, the use of a-tDCS during three days of motor training, also using PPT, produced a significant increase of the unimanual dexterity outcome in all assessed temporal points when is compared to the placebo [58]. Similarly, there was no significant effect on increasing the transference to bimanual fine and gross skills. Other authors found that a-tDCS application during unimanual training improved bimanual motor skills [59]. Specifically, different works found that anodal and classic bi-tDCS during unimanual motor training with PPT improved unimanual and bimanual performance [58,60]. These findings would suggest that tDCS effects are electrode-position dependent.

While non-robust, statistically significant differences were found between groups in the motor dexterity variable, it was interesting to take into consideration the several differences and tendencies that are nearly statistically significant. Regarding the trained task, bi-RP-tDCS seemed to favor the learning retention component. Findings could indicate that bi-RP-tDCS could have effects on memory long-term consolidation, rather than on immediate learning occurring during training. In addition, there is also a tendency to increase the unimanual gross dexterity in the across-sessions component of motor learning and there is a significant improvement in the within-session component in the analysis of the complete sample (*n* = 23), having a possible transference effect for unimanual task but not bimanually as classical bi-tDCS [58,60].

In summary, we interpreted that bi-RP-tDCS did not increase motor learning or its transference, contrary to what had been reported for normal anodal contralateral and classical bi-tDCS, except for having some specific non-significant tendencies for retention. In consequence, these results do not support the cooperative interhemispheric model proposed by Waters et al. [32]. Thereby, the classical competitive model, although reductionist, is able to explain better the small effects reported here, which oppose other works that found an increase in motor performance with a-tDCS [29,58,59,60]. 

In relation to this theory, only one study found that the that bi-hemispheric application improved motor learning, independent of electrode position [32], thus the cooperative model does not have strong support. However, Arias et al. [61] found that both bilateral configurations of tDCS over M1 (bi-tDCS and bi-RP-tDCS) did not change the reaction time in goal-oriented action (arm reaching movements), but reduced the time between the cue and the electromyography (EMG) activation of the deltoid, triceps and biceps brachii muscles (premotor time). Triceps latency time was, instead, increased in the sham group and was associated with fatigability. As the reduction of intracortical inhibition is related with movement initiation, Arias et al. [61] suggested that bi-tDCS could modulate that process, independent of the polarity. Although our results could be better explained by the competitive model, there are other studies and other variables that are not easily explained with this model. Thus, it is still not clear which is the most appropriate model to describe M1 functioning besides the new information that this research had added. 

However, we should highlight other possible reasons to explain the lack of effects of the bi-RP-tDCS in this work. First, a ceiling effect was described in healthy and young people training on motor tasks. They have a good learning capacity that could occlude possible improvements induced by our intervention [16]. However, other works applying a-tDCS over M1 in healthy subjects during one or repeated sessions found an enhanced effect [24,29,59] and, specifically, in fine motor dexterity measured with PPT [18,23,57,62]. This renders the ceiling effect as an unlikely explanation for the lack of effect of bi-RP-tDCS. In relation to this point, although the PPT is a tool commonly used to assess motor dexterity [45,57,63,64], there are many differences in the assessment protocol among studies. For example, instead of counting the number of sticks placed in 30 s, as was originally described [45], some studies registered the total time to complete a full rig with sticks [45,57,64]. In fact, when we analyzed the complete sample, we found a great improvement in the tDCS group over the placebo in the unimanual gross motor tasks (Minnesota) despite it not being the task trained. In contrast with the PPT, the gross motor task measures the time that the participant needed to complete the task, thus, the short time of the PPT task could limit the assessment of small improvements in motor dexterity. Then, we suggest that the way the test was performed could influence the results, through interaction with variables such as fatigue or concentration capacity. Variables that, in fact, can be modulated by tDCS application [24,65]. Finally, the tDCS effect is suggested to depend on task complexity [66]. Our task could be too easy or not motivating enough for subjects due to being repetitive across days.

The secondary objective of this work was to explore the possible grip strength or somatosensory changes produced by bi-RP-tDCS and motor training, and we found no significant effects. The grip strength results were unsurprising, because it is logical to assume that strength increases with specific strength training. Montenegro et al. [67] found that the a-tDCS application in healthy subjects during a maximal strength exercise did not increase knee extension strength compared to the placebo. Cho et al. [68] assessed tDCS on functional recovery of the upper extremity of stroke patients. They concluded that tDCS improved more than the placebo, but the size effect was the same (+3 kg change) in both groups. Both groups had initial differences and, in summary, a-tDCS had no effect over strength training. These results supported that the bi-RP-tDCS application during a dexterity task did not influence grip strength. Regarding the somatosensory variables, one possible explanation for the lack of effects was that our cutaneous somatosensory variables (cutaneous mechanical and pain pressure thresholds, etc.) were different to the sensory variables actually implied in motor performance and learning, such as proprioception or limb position [69,70,71,72]. Nevertheless, other works proposed that somatosensory processing is modulated by M1 stimulation [73], such as a-tDCS effect over M1 increasing somatosensory-evoked potentials [74] or reducing capsaicin-induced hyperalgesia [75]. In this direction, a recent meta-analysis showed that a-tDCS over M1 increased sensory and pain thresholds in healthy subjects [76]. However, other work with the same intervention found no effects in cold and mechanical detection thresholds in healthy subjects, but the cathodal stimulation altered both sensory thresholds [61].

Regarding the limitations of this work, it should be noted that the sample size calculated and applied was done with the main variable, so the results and discussions developed with the rest of the variables should be carefully considered because of the relatively small sample size. Moreover, this study was performed in healthy and young volunteers, thus our results are not generalizable to pathological subjects. 

For future research directions, it would be interesting to compare different types of tDCS applications, not only assessing the behavioral results to determine optimal application, but measuring simultaneously the neurophysiological parameters. For example, transcranial magnetic stimulation (TMS) experimental approaches to measure nervous system excitability or electroencephalography (EEG). These experiments would improve both our understanding of the neurophysiological mechanisms underlying tDCS effect, how the central nervous system works and how motor learning occurs in it. It would also be indispensable to carry out studies in pathological subjects with motor impairments, such as stroke patients, with the ultimate goal of achieving a safe and optimal application of this technique in clinical practice with individualized parametrization.

## 5. Conclusions

Reversed-polarity bi-tDCS application does not facilitate motor learning produced by five days training in healthy subjects, except for a slight tendency towards a learning retention increase in the trained task. Neither affects transferability of the motor learning, grip strength, or somatosensory detection thresholds. The results of this work do not support the brain cooperative model. Although these are better explained by the competitive interhemispheric model, further studies are necessary to unravel tDCS working mechanisms. These results could also add information in order to reach the appropriate method for tDCS study and application.

## Figures and Tables

**Figure 1 brainsci-11-00691-f001:**
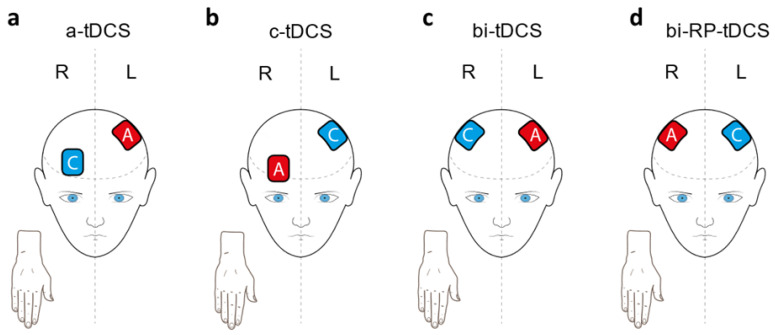
tDCS applications: (**a**) anodal tDCS (a-tDCS) where anode (A) is over contralateral motor cortex (M1) and cathode (C) over homolateral prefrontal cortex; (**b**) cathodal tDCS (C-tDCS) where anode is over homolateral prefrontal cortex and cathode over contralateral M1; (**c**) bi-hemispheric tDCS (bi-tDCS) where anode is over contralateral M1 and cathode over homolateral M1; and (**d**) reversed-polarity bi-hemispheric tDCS (bi-RP-tDCS) where anode is over homolateral M1 and cathode over contralateral M1.

**Figure 2 brainsci-11-00691-f002:**
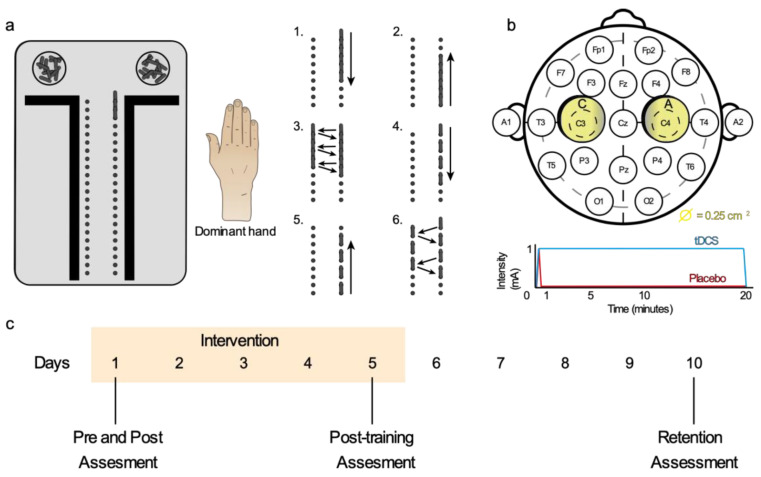
Intervention Methodology. The (**a**) six exercise program with PPT consisted of: 1, placing sticks in one of two vertical columns of holes, one by one, from top to bottom; 2, the same exercise from bottom to top; 3, filling both columns by placing sticks alternately from top to bottom; and 4–6 were exercises 1-3, respectively, but with a free hole left between sticks. After each exercise, participants removed the sticks in reverse order. The (**b**) reversal bi-hemispheric tDCS electrode positioning. We placed the anode on the right (C4) and the cathode on left (C3) primary motor cortex (M1). The (**c**) intervention was applied for five days and the participants’ assessment was performed in basal conditions (previous to any intervention), after the first intervention, after finishing the 5-day intervention, and after 5 days without intervention, to assess retention of motor skills.

**Figure 3 brainsci-11-00691-f003:**
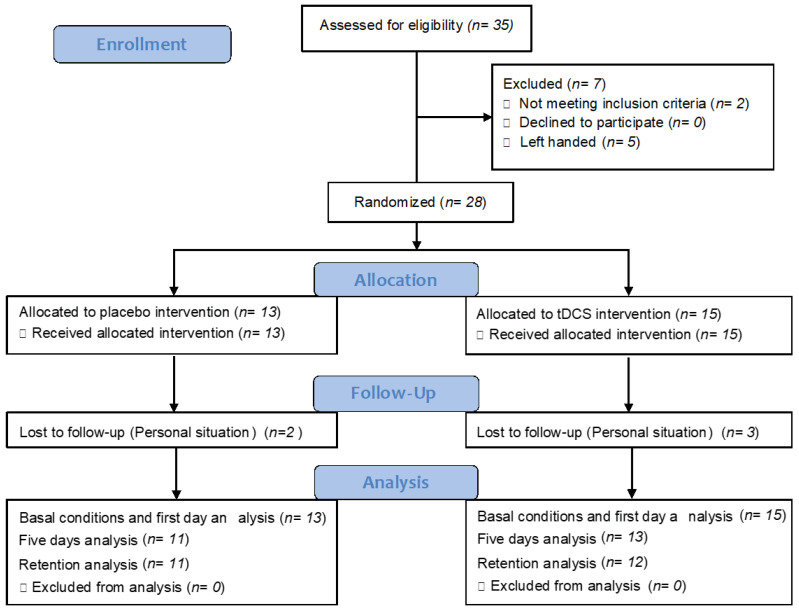
Flow chart of enrollment procedure according to Consort 2010 statement.

**Figure 4 brainsci-11-00691-f004:**
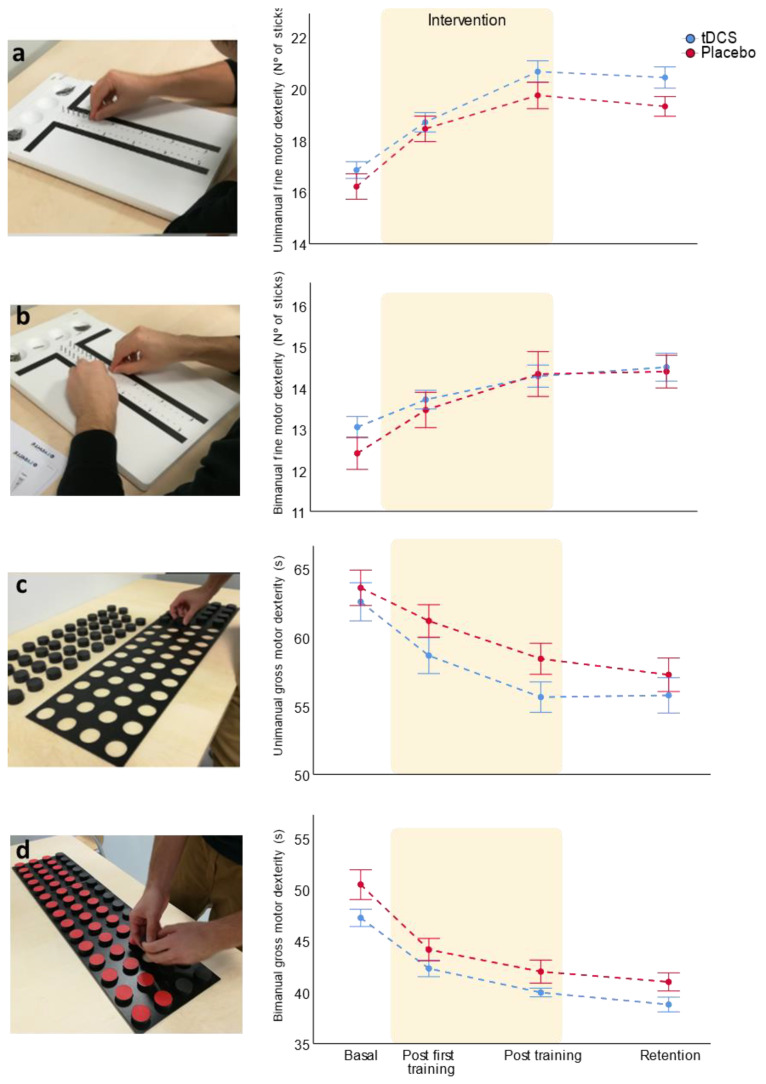
Bi-RP-tDCS effects on motor learning training. This figure shows the temporal evolution of the different motor dexterity variables assessed over days: (**a**) unimanual fine motor dexterity; (**b**) bimanual fine motor dexterity; (**c**) unimanual gross motor dexterity; and (**d**) bimanual gross motor dexterity. The data representation shows mean ± SEM.

**Figure 5 brainsci-11-00691-f005:**
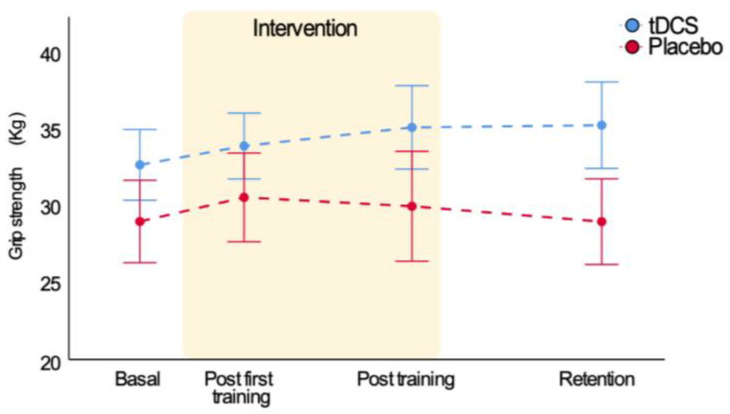
Bi-RP-tDCS effects on grip strength.

**Figure 6 brainsci-11-00691-f006:**
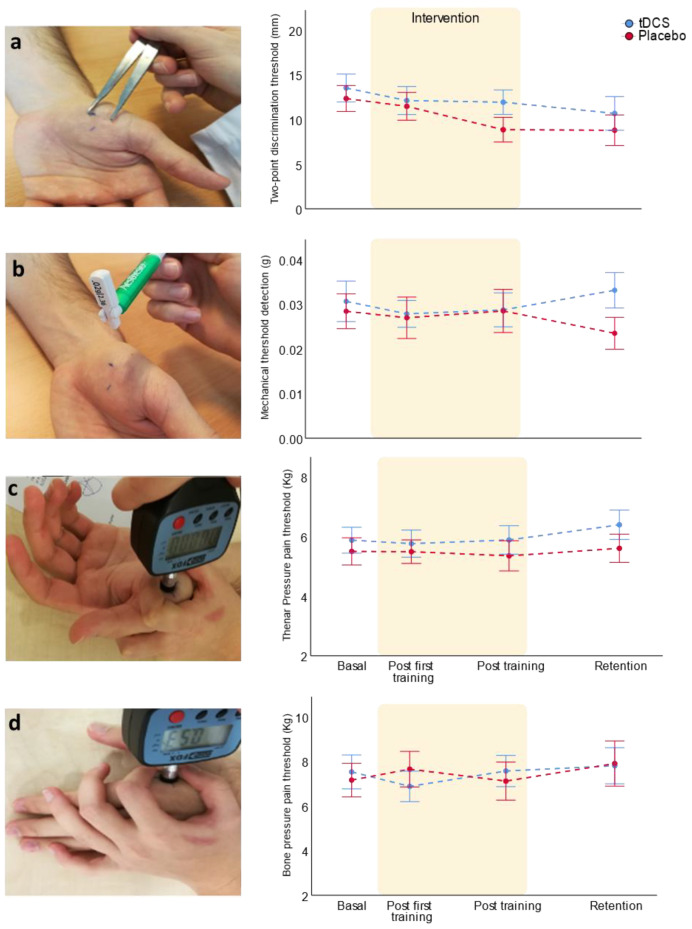
Bi-RP-tDCS and training effects on hand sensitivity over days: (**a**) two-point discrimination; (**b**) mechanical detection threshold; (**c**) pressure pain threshold in the thenar eminence; and (**d**) pain pressure threshold in the diaphysis of the second metacarpal bone. The data representation shows mean ± SEM.

**Table 1 brainsci-11-00691-t001:** Descriptive analysis of sociodemographic data. To evaluate the homogeneity of the different variables between groups. Independent Student’s *t*-test was used for quantitative variables (mean ± standard deviation) and Pearson’s chi^2^ for categorical variables (number of subjects and percentage).

Variables	Bi-RP-tDCS Group (*n* = 15)	Sham Group (*n* = 13)	*p*-Value
**Quantitative**			
Age	26.1 ± 5.5	22.9 ± 1.3	0.006
Sleep Quality	4 ± 2.2	6.9 ± 3.1	0.015
Physical Activity	8962.5 ± 1698.7	7828 ± 1952.6	0.797
Edinburgh Inventory	20 ± 3.7	19.2 ± 3.4	0.570
**Qualitative**			
Gender			
Women	6	5	
Men	9	8	0.256
Educational level			
High school	3	1	
University	12	12	0.353

**Table 2 brainsci-11-00691-t002:** Motor learning during bi-RP-tDCS (mean ± SD).

Motor Dexterity		Group	Basal Condition	Post First Training	Post Training	Post 5 Days without Training	RM ANOVA Factor Time		RM ANOVA Time * Group	
							F	*p*-Value	F	*p*-Value
Fine			*n* = 28	*n* = 28	*n* = 24	*n* = 23		*n* = 23		
	Unimanual	Bi-RP-tDCS Placebo	16.8 ± 1.2 16.2 ± 3.1	18.7 ± 1.5 18.4 ± 1.8	20.6 ± 1.5 19.7 ± 1.7	20.4 ± 1.4 19.3 ± 1.2	84.8	<0.001	2.3	0.085
	Bimanual	Bi-RP-tDCS Placebo	13 ± 1 12.4 ± 1.4	13.7 ± 0.9 13.5 ± 1.5	14.3 ± 1 14.3 ± 1.8	14.5 ± 1.2 14.4 ± 1.3	33.1	<0.001	0.7	0.563
Gross			*n* = 25	*n* = 25	*n* = 21	*n* = 20		*n* = 20		
	Unimanual	Bi-RP-tDCS Placebo	62.6 ± 5 63.6 ± 4.5	58.7 ± 4.7 61.2 ± 4.1	55.6 ± 3.6 58.4 ± 3.6	55.8 ± 4.1 57.3 ± 3.9	41.8	<0.001	2.3	0.089
	Bimanual	Bi-RP-tDCS Placebo	47.2 ± 3 50.4 ± 5	42.3 ± 2.9 44.1 ± 3.8	40 ± 1.4 42 ± 3.5	38.8 ± 2.3 41 ± 2.8	116.8	<0.001	0.1	0.955

**Table 3 brainsci-11-00691-t003:** Motor training and bi-RP-tDCS effects on grip strength (mean ± SD).

Variable	Group	Basal Condition	Post First Training	Post Training	Post 5 Days without Training	RM ANOVA Factor Time		RM ANOVA Time * Group	
						F	*p*-Value	F	*p*-Value
		*n* = 28	*n* = 28	*n* = 24	*n* = 23		*n* = 23		
Grip strength	Bi-RP-tDCS Placebo	32.6 ± 8.9 29 ± 9.6	33.8 ± 8.3 30.5 ± 10.4	35.1 ± 9.8 29.9 ± 11.8	35.2 ± 9.7 28.9 ± 9.2	19	0.007	2.3	0.472

**Table 4 brainsci-11-00691-t004:** Training and bi-RP-tDCS effects on sensory variables (mean ± SD).

Sensory Thresholds	Group	Basal Condition	Post First Training	Post Training	Post 5 Days without Training	RM ANOVA Factor Time		RM ANOVA Time * Group	
						F	*p*-Value	F	*p*-Value
		*n* = 28	*n* = 28	*n* = 24	*n* = 23		*n* = 0.23		
Two-point discrimination	Bi-RP-tDCS Placebo	13.5 ± 6 12.3 ± 5.2	12.1 ± 6.1 11.5 ± 5.6	11.9 ± 4.9 8.8 ± 4.5	10.7 ± 6.5 8.8 ± 5.7	2.6	0.057	0.5	0.681
Mechanical detection	Bi-RP-tDCS Placebo	0.031 ± 0.018 0.028 ± 0.014	.028 ± 0.012 .027 ± 0.016	0.029 ± 0.013 0.028 ± 0.016	0.033 ± 0.013 0.023 ± 0.012	0.5	0.655	1.2	0.319
Thenar pressure pain	Bi-RP-tDCS Placebo	5.9 ± 1.7 5.5 ± 1.6	5.8 ± 1.8 5.5 ± 1.4	5.9 ± 1.7 5.3 ± 1.7	6.4 ± 1.7 5.6 ± 1.6	0.5	0.682	0.2	0.901
Bone pressure pain	Bi-RP-tDCS Placebo	7.5 ± 2.9 7.1 ± 2.7	6.9 ± 2.6 7.6 ± 2.9	7.6 ± 2.5 7.1 ± 2.8	7.8 ± 2.8 7.9 ± 3.4	5.8	0.001	1.7	0.175

## Data Availability

Not applicable.

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
