# Peer review of "Reversed Polarity bi-tDCS over M1 during a Five Days Motor Task Training Did Not Influence Motor Learning. A Triple-Blind Clinical Trial"

_brainsci, 2021, doi:10.3390/brainsci11060691_

Round 1
Reviewer 1 Report
This study investigates the effects of so-called reversed polarity bilateral tDCS on motor training. It’s a very carefully conducted triple-blind study that improves our understanding of the usability of different tDCS setups in a well-established domain. I have rather minor comments/suggestions for the authors but overall enjoyed reading the manuscript.
Abstract:
- It’s misleading to state that 28 subjects participated in the study when there were 5 dropouts. I think you should state the final number here.
Introduction:
- well written, clear logic
- 1: since there’s no consistent way how to depict anode and cathode but anode is often symbolized as +, Fig 1 is somewhat confusing. Please consider naming the electrodes A and C. And also give more information about the setup in the figure caption (e.g., tDCS application over M1 a, anodal tDCS, reference is in place XY…)
Methods:
- All the analyses and tables/figures should only contain the final sample of subjects that finished the study. You can mention the results with all of them in the supplement or as an addition in the text.
- Please review Table 1. The numbers don’t add up or need to be explained better, e.g. gender: 8 males and 8 females in the Sham group = 16!? Also please change the header “significance” to “p” or “p-value”
- Stats:
- have you considered to look at individual changes from baseline as outcome measure after establishing that there are no baseline group differences?
- Is there a subgroup or “responders” that show great improvement?
- If the ANOVA doesn’t reveal a significant interaction, you shouldn’t report ttests per group. It also gets easier to read without these results. The exploratory follow-ups for p<0.1 are justified in my opinion here, though. Mainly because of the rather small sample size. But then it should be clear that these are exploratory.
- Remove # for “almost statistically” significant in Fig 4. It suggests some form of significance of these differences that isn’t justified by the tests.
Discussion:
- Maybe the authors could re-focus on the introduction from which I got the impression that (although there was one study that found a different result), it was never expected to find a tDCS boost on motor learning with this setup. I think this could become clearer in the discussion. That you did find evidence for your hypothesis.
Author Response
First of all, we would like to thank the Editor and the referees for considering our paper and for giving us a portion of their time to revise and contribute to our work. We answered point by point all the required modifications. We hope you will find it adequate and we look forward to your response.
Reviewer 1:
Comments and Suggestions for Authors: This study investigates the effects of so-called reversed polarity bilateral tDCS on motor training. It’s a very carefully conducted triple-blind study that improves our understanding of the usability of different tDCS setups in a well-established domain. I have rather minor comments/suggestions for the authors but overall enjoyed reading the manuscript.
- Thank you very much for that beautiful words and to provide us comments and suggestions to improve our work.
- Abstract: It’s misleading to state that 28 subjects participated in the study when there were 5 dropouts. I think you should state the final number here.
- We completely agree with this suggestion, we have also added the information about the dropouts and the final number (Line 21 and 23): In this randomized, triple-blind clinical trial 28 healthy subjects (14 women) were recruited and randomized into tDCS and placebo groups, although only 23 participants (13 women) finished the complete protocol.
2 Introduction: well written, clear logic.
Since there’s no consistent way how to depict anode and cathode but anode is often symbolized as +, Fig 1 is somewhat confusing. Please consider naming the electrodes A and C. And also give more information about the setup in the figure caption (e.g., tDCS application over M1 a, anodal tDCS, reference is in place XY…)
-We agree with the consideration of naming the electrodes A and C. We have changed Figure 1 and 2b and we have given more information about the setup in the Figure 1 (see in line 107 and 166).
- Methods: All the analyses and tables/figures should only contain the final sample of subjects that finished the study. You can mention the results with all of them in the supplement or as an addition in the text.
- We understand the point. We have included the same tables and figures with n = 23 (excepting the gross motor dexterity, n = 20) in the supplementary material (we attach a Word document). In the text from results we mentioned the suplementary material (line 316-318, 393-396, 415, 431).
- Please review Table 1. The numbers don’t add up or need to be explained better, e.g. gender: 8 males and 8 females in the Sham group = 16!? Also please change the header “significance” to “p” or “p-value”
- Thank you for your observation, we have made a mistake, there are 5 females and 8 males. We also have change the header ‘’significance’’ by ‘’p-value’’ (see in table 1, Line 322).
- Stats: have you considered to look at individual changes from baseline as outcome measure after establishing that there are no baseline group differences?
- Exactly, after establishing that there are no baseline group differences, we have compared the difference between post five days and basal conditions between groups in order to consider the change from the baseline. Regarding more specifically the individual changes, we have plotted the individual subjects progression and we observed that most of subjects had the same tendency.
- Is there a subgroup or “responders” that show great improvement?
- That is a good point, we also asked ourselves that question. We explored the existence of the subgroups of responders, but the participants that showed great improvement belong equally to the tDCS and placebo groups.
- If the ANOVA doesn’t reveal a significant interaction, you shouldn’t report ttests per group. It also gets easier to read without these results. The exploratory follow-ups for p<0.1 are justified in my opinion here, though. Mainly because of the rather small sample size. But then it should be clear that these are exploratory.
- We completely agree with this point. We have eliminated the parts of the text referring to the t-test when the interaction was no significant (see lines 366-367, 380-381, 408-412 and 426-429) Otherwise, we have maintained the exploratory follow-ups for p<0.1.
- Remove # for “almost statistically” significant in Fig 4. It suggests some form of significance of these differences that isn’t justified by the tests.
- We agree with the suggestion. We have removed the # from the Fig. 4 (see line 399).
- Discussion: Maybe the authors could re-focus on the introduction from which got the impression that (although there was one study that found a different result), it was never expected to find a tDCS boost on motor learning with this setup. I think this could become clearer in the discussion. That you did find evidence for your hypothesis.
- Thank you very much, this is a very good point. We have clarified that in lines 442-445: Our hypothesis support that the bi-RP-tDCS, to ensure that its application does not improve motor learning as it does anodal or bi-hemisferic tDCS application with similar parameters [14,28–32], providing evidence following interhemispheric competition model.
The hypotheses supported that the application of bi-RP-tDCS does not improve motor learning as other application modes like anodal or bi-hemispheric tDCS when it is used following similar parameters [14,28–32]. These results provides information in the direction of an interhemispheric competition model.
Reviewer 2 Report
Hello,
Really happy to discover your reserach.
In the abstract, i think you can add retention at the end of the first sentence, because you have worked on that too.
I'm not familiar with somatosensory variables that i have fully understood with pictures. However at the begining it's confuse for me, because these words arrive at the end of the introduction without link before.
Also at the end of the introduction, i think it's more realistic to said that this research give more weight to one model, because we can't better understood the functioning so understand is a bit to strong.
Could you please provide the laterality score in Table 1?
Just to be sure, there is 2.9 difference for sleep quality? Is it possible that this could have an influence on the results?
Here a point of disagreement, you can't compare values with 28, 25,... and 20 participants. I know it's difficult when we have to exclude subjects, but they don't count at the end. So you have to provide informations only of the 20 participants (cf E).
(see 2015 Weissgerber, Tracey L., et al. "Beyond bar and line graphs: time for a new data presentation paradigm." PLoS Biol.)
Also i need to have effect size especially since you speak about tendency in the discussion (cf paper attached).
Perhaps using linear mixed model could you provide better information. Indeed when we compare the F value for Time and Timexgroup, mixed model is a way to have a better view of the data.
I think you could provide more development in the discussion with the grip strength that could improve the manuscript.
Finally, i'm not sure to well understood the last sentence. You said parameters, but you can just provide informations on the montage which is already a step foward.
Best,
Author Response
First of all, we would like to thank the Editor and the referees for considering our paper and for giving us a portion of their time to revise and contribute to our work. We answered point by point all the required modifications. We hope you will find it adequate and we look forward to your response.
Reviewer 2
Comments and Suggestions for Authors: Hello, Really happy to discover your research.
- Thank you very much for your words and your really useful suggestions that contribute to improve the manuscript.
- In the abstract, i think you can add retention at the end of the first sentence, because you have worked on that too.
- We completely agree. We have added in line 18: Our purpose was to explore the reversal bilateral tDCS effects on a manual dexterity training, during one, five days and the retention component measured after XXX days, and whether somatosensory effects were produced
- I'm not familiar with somatosensory variables that i have fully understood with pictures. However at the begining it's confuse for me, because these words arrive at the end of the introduction without link before.
- Thank you for that point, we agree with you, we have missed a link in the introduction. We have added in line 52: Due to the close relation between motor and sensory cortex, motor learning has an effect also on sensory perception, producing changes related with somatosensory input or movement perception [12].
- Also at the end of the introduction, i think it's more realistic to said that this research give more weight to one model, because we can't better understood the functioning so understand is a bit to strong.
- We have eliminated ‘’but also add more basis to understand which theory model is more adecuate’’. Finally we have written ‘’This work could clarify whether the bi-RP-tDCS enhances motor learning or not, giving more weight to the competitive model, that could be more adequate to explain brain performance (Line 102-103).
- Could you please provide the laterality score in Table 1?
- Of course. We have added the laterality score (Edinburgh Inventory) in Table 1 (Line 322).
- Just to be sure, there is 2.9 difference for sleep quality? Is it possible that this could have an influence on the results?
- Exactly, there is a difference of sleep quality between groups regarding the mean of the Pittsburg score. To score more than 5 points is considered as a ``bad sleeper´´, then the placebo groups are worse sleepers than tDCS. Thus, we have explored if this variable could influence on the results. We have performed a linear regression and the sleep did not predict the motor learning (variable used: difference between post 5 days training and basal condition) (R2 = 0.074, p = 0.232). Also, we have compared the proportion of bad sleepers between groups and there are not differences (p = 0.123). We have added this explanation to the discusión (Line 523).
- Here a point of disagreement, you can't compare values with 28, 25,... and 20 participants. I know it's difficult when we have to exclude subjects, but they don't count at the end. So you have to provide informations only of the 20 participants (cf E). (see 2015 Weissgerber, Tracey L., et al. "Beyond bar and line graphs: time for a new data presentation paradigm." PLoS Biol.)
- We perfectly understand the disagreement. However, if we have 25 subjects that completed the motor training, then, we think that is ethic to report that data in order to answer the specific hypothesis about how tDCS influenced the motortraining. Thus, and as the other reviewer has suggested, we have provided the data information and representation of the participant that completed all measurements at supplementary material and we have referenced it in the text (See lines 316-318, 393-396, 415, 431).
- Also i need to have effect size especially since you speak about tendency in the discussion (cf paper attached).
- We completely agree with you. We have reported the Cohen’s d in the cases which we have spoke about tendency in the results (See line 351, 384 and 389)
- Perhaps using linear mixed model could you provide better information. Indeed when we compare the F value for Time and Timexgroup, mixed model is a way to have a better view of the data.
- Thank you for the approach suggestion. We have added in the supplementary material the linear mixed model, reporting the F and p values for Time, Group and Timexgroup. (See the document attached: Supplementary File)
- I think you could provide more development in the discussion with the grip strength that could improve the manuscript.
Of course, thank you very much for your observation. We have added the next argumentation: Montenegro R. et. al. [67] found that the a-tDCS application in healthy subjects during a maximal strength exercise did not increased the knee extension strength compared to placebo. Besides, Hyuk-Shin Cho, et. al. [68] assess tDCS on functional recovery of the upper extremity of stroke patients. They conclude that tDCS improved more than placebobut the size effect is the same (+3 kg change) in both groups. Both groups have initial differences and we can summarize that the a-tDCS has no effects over strength training. These results supportted that the bi-RP-tDCS application during a dexterity task did not influence grip strength (See line 534)
- Finally, i'm not sure to well understood the last sentence. You said parameters, but you can just provide informations on the montage which is already a step foward.
We agree with you. We have changed ‘’parameters’’ by ‘montage’’ (See line 578).
Lastly, we have detected an error in the references, then we have revised and corrected the bibliography.